# The Carol about the Pagan Rite of Sacrifice of a Goat and Its Interpretation in Russian Scholarship of the 19th to 20th Centuries

**Andrey Toporkov** 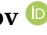

Gorky Institute of the World Literature, Russian Academy of Sciences, 121069 Moscow, Russia; atoporkov@mail.ru

**Abstract:** In publications of Russian folklore, along with authentic texts there are a number of literary stylizations based on folklore. The article traces the history of one such pseudo-folkloric text—a carol which was first published by Ivan Petrovich Sakharov (1807 to 1863) in 1837. It has been established that this carol is a montage of two texts: the first is a carol, printed in 1817 by I.E. Sreznevsky in the *Ukrainian Bulletin*, and the second is a song included in the *Tale of Brother Ivanushka and his Sister Alyonushka* (SUS 450). Such contamination is unique and is found only in this one text, which was later reprinted many times. Taking into account Sakharov's reputation as a falsifier of folklore, there is no reason to doubt that it was he who composed this carol; such contamination of works belonging to different folkloric genres is also characteristic of other of Sakharov's publications. The carol that Sakharov published attracted the particular interest of researchers of Slavic mythology due to the fact that it described how an old man was going to sacrifice a goat. Several generations of historians saw in this pseudo-folkloric text a description of a ritual that pagan Slavs performed in ancient times. Considering the carol as an historical document, researchers of mythology built their interpretations based on the supposed time of its appearance, the nature of its genre, plot, and individual details. Thus, Sakharov's pseudo-folkloric creation found an eager audience among scholars, and it stimulated their imagination in picturing the life of pagan Rus'.

**Keywords:** folklore; Ivan Petrovich Sakharov; literary pastiche; a hoax; a fake; Christmas carols; the sacrifice of a goat

## 1. Introduction: I. P. Sakharov as Publisher of Folklore

Researchers of the pre-Christian culture of ancient Rus from the late 1830s to almost our time have regarded I. P. Sakharov's carol ("Za rekoiu, za bystroiu . . . " [Beyond the river, beyond the swift]) as evidence of the pagan ritual of animal sacrifice. Sakharov first published the carol in 1837, and it was reprinted the following year by I.M. Snegirev. It was then reproduced in many publications and was accepted by many scholars of Slavic mythology. However, the evidence indicates that this work was Sakharov's own creation, adapted from carols and other material from folklore. This article examines the origins of this falsification and its subsequent history.

Ivan Petrovich Sakharov (1807 to 1863), a local historian, paleographer, archaeologist, and publisher of historical and folkloric materials, occupies a special place in the history of Russian culture. Sakharov came from the clergy; his father, a priest, died when the boy was six years old. In 1815, Sakharov entered the Tula district religious school, and in 1822, the Tula seminary. After graduating in 1830, Sakharov went to Moscow, where he entered the medical faculty of Moscow University. Upon graduation, Sakharov first worked as a doctor in Moscow, and in February 1836 moved to St. Petersburg. For many years, Sakharov remained a practicing physician, devoting all his free time to studying folklore and the book tradition.

In 1835, Sakharov prepared the first part of his *Tales of the Russian People About the Family Life of Their Ancestors*, which was published in 1836, after he moved to St. Petersburg. A second edition came out in 1837, and in the same year, the second and third parts of *Tales of the Russian People* were published. After that, Sakharov began preparing a new expanded edition, but his plan was only partially implemented. Instead of the seven volumes in 30 parts that he had conceived, two volumes in eight parts were published. In 1838 to 1839, Sakharov also published a collection of *Songs of the Russian People* in five parts.

In the 1830s to 1840s, the distinction between publications of folklore and works of art on themes from folk life was not yet clearly understood. The lines between a story from folk life, a "physiological" sketch and an ethnographic account were blurred. Several Russian and Ukrainian writers of the time collected and published folklore and at the same time used it in their own artistic works. In journals intended for a wide audience, pictures of folk life were fictionalized and generously embellished with fantasy.

Sakharov had his own publishing house; it published numerous books and may be considered a successful commercial undertaking. Judging by contemporaries' admiring reviews, his publications in which folklore was stylized in a pseudo-folk spirit were popular with the reading public. Ideologically, Sakharov adhered to the doctrine of "Official Nationality" and his appeal to ancient legends, fairytales, songs, icon painting, and church singing was motivated not only by the desire to preserve them for posterity, but also to counteract the values of Western civilization.

Folklorists, ethnographers, and literary historians have confirmed that Sakharov composed fairytales which he passed off as authentic, using the plots of genuine *byliny* and referring to non-existent manuscripts. He also edited the texts of charms (*zagovory*) and riddles and made insertions in the texts of ancient Russian monuments. Although Sakharov claimed to publish mostly his own recordings of folklore and asserted that he reproduced them with great accuracy, doubts about the reliability of his materials were expressed, even during his lifetime.

Nevertheless, collections of folklore materials prepared by Sakharov continue to be republished. It should be borne in mind that not everything in them should be considered "fakelore". He took some texts from existing editions of folklore. Many collectors of the time also gave materials to Sakharov. The letters of the Tikhvin merchant G.I. Parikhin, who sent Sakharov the texts of riddles, songs and charms, have been preserved. On the basis of this correspondence, one can gain an idea of Sakharov's relationship with his local correspondents and of how he reworked the original materials he published (Toporkov 2014). Sakharov made extensive use of a kind of montage in which fragments of authentic texts were combined with sections that he himself composed. In Sakharov's editions, there are texts that are completely authentic as well as those that underwent only slight stylistic editing. At the same time, there are also texts entirely composed by him, albeit with subject matter taken from folklore. The pseudo-folkloric texts that Sakharov composed were subsequently reprinted in other publications; as is the case with "Za rekoiu, za bystroiu . . . ", scholars have used them to reconstruct pagan mythology, so poorly represented in authentic historical sources.

For evaluating Sakharov's publishing activity, the history of the perception of his texts in the subsequent tradition is thus no less important than the history of their first appearance. In this article, we will try to answer two main questions. Firstly, what is the origin of Sakharov's carol—what comes from folklore and what does not? Secondly, how was the carol perceived by readers and how did scholars of Slavic mythology regard it in the 19th and 20th centuries?

## 2. The Origins of Sakharov's Carol

Sakharov published the carol under consideration three times. The first was in the second part of the first edition of *Tales of the Russian People* in the section "Russian Family Songs" (Sakharov 1837, pp. 257–58, no. 63). The second was in the first part of *Songs of the Russian People*, in the section "Carols" (*Pesni koliadskie*) (Sakharov 1838, pp. 94–95, no. 2),

and the third time in the third book of the first volume of *Tales of the Russian People*, also in the section "Carols" (Sakharov 1841, p. 16, no. 2; this edition is labeled "third" on the cover, although in fact it is not). The first publication of the carol was printed without comment, while the second and third publications gave the place of alleged recording and cited another carol, published in 1817 by I.E. Sreznevsky (the father of the famous Slavist), as a parallel.

Here is the text from the publication in the first part of *Songs of the Russian People* (Sakharov 1838):

> За рекою, за быстрою,
> Ой колiодка! ой колiодка!
> Леса стоят дремучие,
> Во тех лесахогни горят,
> Огни горят великие,
> 5 Вокругогней скамьи стоят,
> Скамьи стоят дубовыя,
> На тех скамьях добры молодцы,
> Добры молодцы, красны девицы,
> Поют песни калiодушки.
> Ой колiодка! ой колiодка!
> 10 В средине их старик сидит,
> Он точит свой булатной нож.
> Котел кипит горючий,
> Возле котла козел стоит;
> Хотят козла зарезати.
> Ой колiодка! ой колiодка!
> 15 Ты, братец, Иванушко,
> Ты выди, ты выпрыгни!
> Я рад бы выпригнул(так!),
> Горюч камень
> К котлу тянет,
> 20 Желты пески
> Сердце высосали.
> Ой колiодка! Ой колiодка!

(Sakharov 1838, pp. 94–95, № 2)

[Across the fast river, / Oh, *koliodka*! Oh, *koliodka*! / Bright lights are burning, / Benches stand around the lights, / Oak benches stand, / On those benches there are good young people, / Young people [and] beautiful girls, / They sing songs to the *kaliodushka*. / Oh, *koliodka*! Oh, *koliodka*! / An old man sits among them, / He sharpens his steel knife, / The cauldron is boiling hot, / Near the cauldron a goat stands, / They want to slaughter the goat. / Oh, *koliodka*! Oh, *koliodka*! / You, brother Ivanushka, / Come out, jump out! / I'd love to jump out, / [However] the burning stone / Pulls [me down] into the cauldron, / Yellow sands sucked out [my] heart. / Oh, *koliodka*! Oh, *koliodka*!]

In numbering the lines, we did not count the chorus ("*Oi koliodka! Oi koliodka!*"). Judging by Sakharov's publication, the chorus was to be performed four times; choruses are often repeated after each verse when performing ritual songs. Such exclamatory refrains are found in Russian Christmas songs, which are usually called "carols" (*koliadki*). In these same songs "*koleda*" (or "*koliada*", "*koliadushka*", "*kaliodushka*", "*koliodka*", also meaning "carol") is personified and her actions are depicted. For example, a carol recorded in the village of Vasilevo, Dmitrovsky district, Moscow province, begins with the words:

> *Koljoda, Koljoda!*
> *Koleda* came

To the master's yard
On Christmas Eve . . .

([Shejn 1898](), p. 306, no. 1035)

It is noted that "boys call to *koleda* on the evening before Christmas" ([Shejn 1898](), p. 306, no. 1035). Similarly, a carol from the village of Storozhi, Efremovsky district, Tula province, begins:

*Koljodushka-koleda*!
*Koleda* walked about
On Holy Evenings.

([Shejn 1898](), p. 306, no. 1033)

Sakharov notes that the carol was recorded "from the words of Tula villagers" ([Sakharov 1838](), p. 163, note 18). In the section "Variants of Christmastide Songs", he claims that only one version of the song exists, the one printed by Sreznevsky in the *Ukranian Bulletin* in 1817:

За рекою, за быстрою, ой каліодка!
Леса стоят дремучие,
Во тех лесахогни горят,
Огни горят великие,
5 Вокругогней скамьи стоят,
Скамьи стоят дубовыя;
На тех скамьях добры молодцы,
Добры молодцы, красны девицы
Поют песни каліодушке;
10 В средине их старик сидит,
Он точит свой булатной нож;
Возле его козел стоит.

[Beyond the river, beyond the fast one, oh *kaliodka*! / The dense forests stand. / In those forests fires are burning, / Great fires are burning, / Benches stand around the fires, / Oak benches stand; / On those benches good fellows, pretty girls / Sing songs to *kaliodushka*; / In the middle of them an old man sits, / He sharpens his steel knife; / A goat stands beside him.] ([Sakharov 1838](), pp. 129–30)

Sakharov states that "Something is missing from this ritual song. The song in our collection fills in all the omissions ..."

In Sreznevsky's article "Slavic Mythology, or About Russian Pagan Worship" of 1817, Sakharov's carol is cited in the context of pagan myths and rituals:

Koliada, according to some, was the Slavic god of peace and corresponded to the Roman Janus. His feast was celebrated on December 24, which was celebrated with celebrations and feasts. The temple of this god (or perhaps goddess) was in Kiev; we do not know anything about how it looked.

In some places, superstitious people still celebrate this day with almost the same rituals as the ancients. As a child, I happened to hear one hymn to Koliada, of which I can only remember the beginning; here it is [. . . ]. This beginning demonstrates the rite of ancient sacrifice itself.

([Sreznevskij 1817](), pp. 19–20)

Thus, Sakharov's assertion that "Something is missing in this ritual song" is based not only on acquaintance with the song itself, but also on Sreznevsky's direct testimony that he could only remember its beginning.

As a child, Ivan Evseevich Sreznevsky (1770 to 1819) lived in the village of Sreznevo, Spassky district, Ryazan province, and then studied at the Ryazan religious school. This means that, most likely, he heard the carol that he recited from memory in Ryazan or its environs. Sakharov's statements that the text was written "from the words of Tula villagers"

and that "Something is missing in this ritual song. The song in our collection fills in all the omissions" are to some extent contradictory, and also at odds with the fact that the first part of Sakharov's carol is almost identical to that of Sreznevsky. It is difficult to imagine that the "Tula villagers" really knew a text that, on the one hand, almost literally repeats the carol published in 1817, and on the other, "fills in all the omissions" of the carol. If such a text were really performed orally, it would inevitably have contained some differences from the carol that Sreznevsky heard in childhood (probably in Ryazan province). It is much easier to imagine that Sakharov himself took the carol Sreznevsky published and added to it. He was familiar with the publication in the *Ukrainian Bulletin* and modified other texts of the oral tradition, such as those collected by Parikhin (Toporkov 2014).

The first 11 lines of the carols of Sakharov and Sreznevsky coincide. Then, Sakharov has as the 12th line, absent in Sreznevsky, "The cauldron is boiling hot". Verse 12 in Sreznevsky's text is: "A goat stands beside him", which in Sakharov corresponds to the 13th verse: "Near the cauldron the goat stands" with the replacement of the pronoun "him" by the noun "cauldron". Note that the word "cauldron", which appears twice in Sakharov's carol, does not occur at all in Sreznevsky's. The next eight verses are found only in Sakharov's text.

The question also inevitably arises whether we can trust Sreznevsky. Is his version not also a fake? To answer this question, editions of authentic folklore offer guidance. Three carols, two from the Orenburg province and one from the Omsk region, are quite close to Sreznevsky's. The first carol was recorded in the village Podgornaya Pokrovka in the Orenburg district (Celebrating Christmas carols 1888). The second was written down by A.P. Kuznetsov in the same Orenburg province (without specifying the place Shejn 1898, p. 309, No. 1046; reprint: Zemtsovskij 1970, p. 71, No. 20; see ibid. p. 550). The carol was also recorded by N. Kravets in 1975 in the village of Loginovka, Pavlogradsky District, Omsk Region (Bolonev et al. 1997, p. 51, No. 1). In all three carols, there are steep mountains, fast rivers, and dense forests; people are standing around burning fires, but there is no old man, no goat, no steel knife, and people are not going to kill anything but are just singing carols.

There are also parallels to the carols under consideration in Belarusian ritual poetry; however, there is no theme of sacrifice there either. For example, in a carol from the town of Molodusha Rechitskiy in Minsk province, fires and cauldrons are also described, although they are not located in the forest but in the courtyard of the owner of the house. Wine, beer and "*koledka*" are brewed in cauldrons (Shejn 1887, p. 76, no. 67; reprint: Mazhejka 1975, p. 220, no. 276). In ritual songs with a similar plot, images of grandfathers sometimes appear. For example, in the Belarusian Easter song (*volochebnaia pesnia*), old men melt wax for candles in cauldrons (Bartashevich and Salavej 1980, p. 209, no. 116). The motif of the impending sacrifice of a goat has parallels in the Ukrainian and Belarusian rituals of "leading a goat". These Christmas rituals stage the killing of an animal which then miraculously comes back to life.[1] In some of the songs accompanying the ceremony, the goat says that it is afraid of its old grandfather (Romanov 1912, p. 105).

Thus, motifs that are found in Sreznevsky's text have parallels in three Russian carols and in Belarusian carols and Easter songs. The motif of killing a goat, which is only suggested in Sreznevsky's text, is absent in the other carols; however, as noted, it does appear in the Belarusian and Ukrainian ritual of "leading a goat". Hence, we may state that Sreznevsky's carol fits quite organically into the Christmastide ritual poetry of the Eastern Slavs.

Despite the coincidence of Sakharov and Sreznevsky's texts, the former's edits are not a reworking of the original. Instead, Sakharov amends the texts with elements of a fairy tale. In one song from a fairytale, "a brother, walking with his sister on the road, drinks water from under a goat's hoof and turns into a kid; the sister is getting married, but the witch drowns her and replaces her (with herself or with her daughter), and they want to slaughter the kid; [but] everything is then revealed" (Barag et al. 1979, p. 135, no. 450).

The contamination of a carol with a song from a fairytale is unique and only occurs in Sakharov's text.

In the earliest known version of this fairytale, published in the collection *An Old Tune in a New Setting* (1795), a boy who has turned into a lamb addresses his sister with the words: "Sister! Alyonushka! Come here, say goodbye to me: they are sharpening steel knives, cast-iron cauldrons are boiling, they want to kill me, a lamb!" And his sister answers him from under the water: "Oh, my brother Ivanushka! I would be glad to come out to you, [but] spring water is washing into my eyes, a burning stone pulls me to the bottom!" (An Old Tune in a New Setting 2003, p. 130, no. 12). In some later published versions of the tale, the brother and sister's dialogue grows in length, and is graphically presented like poetry. Sometimes it is stipulated that the performers must sing these lines. One of the most extensive versions reads:

> Попросился козлёночек перед смертью к воде, напиться, пришел на бережок и плачет:
>
> Алёнушка,
> Сестрица моя!
> Выплынь, выплынь,
> На бе́режок,
> Ты выдь ко мне,
> Промолвь со мной:
> Костры кладут
> Высокие,
> Котлы висят
> Глубокие,
> Огни горят
> Горючие,
> Смолы кипят,
> Кипучия,
> Ножи точа́т
> Булатные,
> Хотят меня
> Зарезати!
>
> А сестрица Алёнушка отвечает ему со дна:
>
> Ты братец мой,
> Иванушка,
> Иванушка,
> Козлёночек!
> Я рада бы
> Помо́чь тебе,
> Тебе тошно́,
> А мне тошней:
> Тяжол камень
> Ко дну тянет;
> Шелкова́ трава
> На руках свила́сь,
> Ноги спутала;
> Желты пески
> На грудь легли;
> Люта́ змея
> Сердце высосала;
> Бела́ рыба
> Глаза выела!

Погорюет, погорюет козлёночек, а как видит себе скорую смерть, до трех раз просился к воде и звал сестрицу. ([Bessonov 1868](), pp. 122–24)

[Before dying, the little goat asked for water to drink, came to the shore, and cries:

Alyonushka, / My sister! / Swim out, swim out, / Onto the shore. / Come out to me, / Say with me: / Bonfires are made, / Tall ones, / Cauldrons hang / Deep ones, / They burn / Hot, / The pitch boils / Boiling, / The knives are sharpened / Of steel, / They want / To slaughter me!

And his dear sister Alyonushka answers him from the deep:

You, my dear brother / Ivanushka, / Little goat! / I would be glad / To help you. / You feel sick, / But I am sicker: / A heavy stone / Is pulling me to the bottom;/ Silken grass encircled my arms, / It caught my legs, / Yellow sands / Lay down on my chest; / A fierce snake / Sucked out my heart, / A white fish / Ate out my eyes!

The little goat grieves and grieves, and as he sees death approaching, he asked to go up to the water up to three times and called his sister.]

The fourteenth line from Sakharov's carol ("They want to slaughter a goat") has a parallel in a fairytale from Saratov province published by A.N. Afanasyev: "The little goat walks to the riverbank and cries bitterly, saying: "Olenushka, my dear sister! Come out to me, take a look: I am your brother, Ivanushka, I have come to you with unhappy news, they want to kill me, [a goat], to slaughter me ..." Olenushka answers him from under the water: "Oh, my brother Ivanushka! I would be glad to look out at you, [but] a heavy stone is pulling me to the bottom ..." ([Afanas'yev 1985](), p. 255, no. 263).

In verses 15 to 21, Alyonushka's response to her brother is paraphrased, which is directly indicated in line 15: "You, brother, Ivanushko . . . ". The words from the carol "You go out, you jump out!" in the fairytale quoted above from Afanasyev's collection correspond to: "You come out to me, you look out ..."—but this is Ivanushka's remark addressed to Alyonushka, and not vice versa, as in Sakharov's text.

In the fairytale, Ivanushka asks his sister to look out from under the water, and she explains why she cannot, while in Sakharov's carol, Alyonushka asks Ivanushka to come out from somewhere and to jump out. However, this is not motivated, since neither Ivanushka nor Alyonushka are located in the water. Sakharov's text is strange: "A burning stone // Pulls [me down] into the cauldron" instead of "to the bottom". By the way, V. Ia. Propp, quoting the carol, replaced "pulls into the boiler" by "pulls to the bottom" and noted: "[This is] corrected from an obvious error" ([Propp 1995](), p. 57, note 15). In this case, however, we are dealing not with an "obvious error" but with a deliberate change to the original text. Several semantic inconsistencies also indicate the contamination of texts of different genres.

If in Sreznevsky's carol there was only a hint of a possible future killing of a goat, in Sakharov's text this turns into a whole scene of preparation for the murder of a child who has been turned into a kid, complete with dramatic remarks from the proposed victim. In this context, the killing of a goat is on a par with human sacrifice, and the whole situation takes on a menacing character that is not at all characteristic of carols. The motif of killing a goat is not found in carols, but in songs accompanying the Christmas ritual of driving a goat to sacrifice.

Thus, Sakharov's carol is clearly contaminated and shows inconsistencies in the dialogue between Ivanushka and his sister. However, despite the eclecticism and imperfection of the text, it has repeatedly been reprinted and taken seriously in studies on Slavic mythology.

### 3. Subsequent Use of Sakharov's Carol as Historical Evidence

Ivan Mikhailovich Snegirev was the first to reprint Sakharov's carol in the section "Carols and Vinograd'e Songs" in the second part of his *Russian Folk Holidays and Rites* (Snegirev 1838, pp. 68–69, no. 4). In a note to the song Snegirev describes its contents as follows: "Here is depicted a sacrifice in which sand is poured, into which among Northern peoples the blood of the victim was spilled; this is expressed by the words the sands have sucked out the heart. See the song in the first part of this book, p. 103" (Snegirev 1838, p. 69, note 1). Snegirev thus refers the reader to the first issue of *Russian Folk Holidays and Rituals*, where first the sacrifice of a goat by Lithuanians is described and then Sreznevsky's carol is cited as evidence that such rituals were known in Russia: "Observation reveals that this celebration with more or less similar rituals and sacrifices was common not only throughout Lithuania, but also in Lithuanian Rus. According to the testimony of Mr. Sreznevsky, in southern Russia they sang the following song, which depicts a sacrificial ritual similar to the Lithuanian one" (Snegirev 1837, p. 103).

In his book *Slavic* Mythology (1847), Nikolai Ivanovich Kostomarov wrote with reference to Sakharov's material that during the winter holidays "sacred games were performed, as can be seen from winter round dances, and sacrifices were made. Thus, in one Russian Christmas song, an obviously pagan sacrifice is portrayed: fires are burning in the forest, benches are standing around them, young men and women are sitting on benches" (Kostomarov 1847, pp. 100–1).

Izmail Ivanovich Sreznevsky referred to our carol in his book *Studies on the Pagan Worship of the* Ancient Slavs (1848). He cited the text of the carol in a review of materials on pagan sacrifice: "There is a legend about the sacrifice of a goat preserved in a folk ritual song: 'Beyond the river [...] They want to slaughter a goat.' According to the testimony of Dlugosh, the Poles also made sacrifices: sheep and bulls were sacrificed during festivities at which people gathered. They also occurred among the Baltic Slavs" (Sreznevskij 1848, p. 73).

The historian Sergei Mikhailovich Soloviev, in his "Sketch of the Customs, Rites and Religion of the Slavs, Mainly the Eastern, in Pagan Times" (1850), noted that "among the carols the following is remarkable" and quoted Sakharov's text (Solov'ev 1850, p. 30). He further considered the ritual sacrifice of a goat in comparative context (Solov'ev 1850, p. 31).

Fyodor Ivanovich Buslaev brought up Sakharov's carol in one of his lectures on the history of Russian literature that were read to the heir, Tsarevich Nikolai Alexandrovich, in 1859 to 1860. Speaking about Christmastide rites, Buslaev noted: "It goes without saying that in carols the Nativity of Christ is recalled, but beyond the Christian element, the most ancient pagan material is also evident. Especially important in the latter respect is the carol about the sacrifice of a goat, which has survived to this day in Southern Russia ..." (Buslaev 1990, p. 434). Having quoted Sakharov's text in full, Buslaev continued: "We already know that the goat was dedicated to Toor-Perun, the deity of agriculture and family settlement. He rode in a cart pulled by two goats" (Buslaev 1990, p. 435).

In the article "Kupala and Kolyada in Relation to the Folk Life of the Russian Slavs", included in his book *Russian Nationality in its Beliefs*, Rituals and Fairytales (1862), Dmitry Ottovich Shepping quoted Sakharov's text, making some adjustments. Most importantly, he considered the text not a carol, but a *kupala* (summer solstice) song, and in this connection he replaced the line "Singing songs of *Kolyudushka*" with "Singing songs of *Kupalushka*" (Shepping 1862, p. 40). Shepping associated Ivanushka's name with the night of Ivan Kupala: "That this song does not belong to the carols, but on the contrary, to the celebrations of Ivanovka night, is clearly indicated by the name of the goat Ivanushka, which is why we consider ourselves entitled to replace the word *Kolyudushka* with the word *Kupalushka*" (Shepping 1862, p. 40, note 1). In fact, the name of Ivanushka in the carol is borrowed from a fairytale and has nothing to do with Ivan Kupala.

According to Shepping, Sakharov's song testified that in antiquity some living creatures were burned in the Kupala bonfire: "It is likely that during pagan times live sacrifices were also made to the Bathing Fire (*Kupal'nomu ogniu*), as we see from this ancient song . . . "

(Shepping 1862, p. 39). Shepping identified the old man who appears in the carol with a pagan priest: "The very production of living fire, as in the lighting of Ivanovo bonfires, is usually entrusted to old people, who, as can be seen from this rite—one of a purely communal life, and from the above mentioned song, performed the office of a priest in the public worship of pagan Russia" (Shepping 1862, pp. 40–41).

Alexander Nikolaevich Afanasyev, in his three-volume study *Poetic Views of the Slavs on Nature*, noted that: "A remembrance of the sacrificial slaughter of a goat is preserved in a carol, which is all the more curious because it also conveys the very setting of the ceremony . . . " (Afanas'yev 1868, pp. 257–58). Citing Sakharov's text, Afanasyev continues: "The sacrifice was accompanied by the singing of ritual songs; the goat was cut by the elder and its meat was cooked in a cauldron, just as was done among the Germanic tribes and the Scythians. Among the Lithuanians, the priest, stabbing a goat with a sharp knife, invoked God's blessing on those praying; blood was collected in a jug or bowl, and then people, cattle and even dwellings were sprinkled with it; the meat was eaten while singing songs and trumpet playing, and what remained uneaten was buried in the ground at the crossroads so that not a single animal could touch the sacred food" (Afanas'yev 1868, p. 258).

The 19th-century tradition of mythological research was revived in the 1960s to 1980s by the archaeologist and academician Boris Alexandrovich Rybakov. In his monograph *The Paganism of Ancient Rus* (1st ed.—1987), he cited the carol from Snegirev's collection and noted that this text "reveals the essence of the ritual ceremony—the sacrifice of a goat . . . " (Rybakov 1987, p. 85). Rybakov, following Shepping, wrote that the setting of the sacrifice is reminiscent of the Kupala holidays: "The name of brother Ivanushka may indicate a ceremony on the night of Ivan Kupala; then sister Alyonushka is Kupala herself, a victim doomed to become 'engulfed in water.' On the night of Kupala, 'great fires burn,' and rituals are performed near the water, imitating the drowning of the victim: bathing a girl dressed up as Kupala or immersing a stuffed doll depicting Kupala in the water" (Rybakov 1987, p. 95). It is hard to agree with this reasoning. As we have already noted, the name Ivanushka is borrowed from a fairytale and has nothing to do with the night of Ivan Kupala; Alyonushka is identified with the mythical Kupala only in Rybakov's imagination.

## 4. Conclusions

Thus, researchers have regarded Sakharov's carol "Za rekoiu, za bystroiu . . . " as authentic evidence of pagan rituals of pre-Christian Rus. In order for the carol to be seen as an historical document, they have interpreted its generic nature, time of appearance, plot, and individual details in a particular way. First, the song was viewed not as a traditional carol, performed to congratulate homeowners on the feast of the Nativity of Christ, but as a miraculously preserved description of a real ritual practiced in ancient Russia, as if it were a fragment of a chronicle or a work by an Arab or Byzantine author of the tenth century. Second, scholars have seen in the work a description of the sacrifice of a goat, although Sreznevsky's text speaks more about preparing a goat for slaughter, and Sakharov's carol— about the imminent killing of child who has been changed into a goat. The modern reader sees in the refrain "Oh, *koledka*!" an appeal to the holiday of *Kolyada*, as Christmas Eve was called, contrary to Sreznevsky's assertion that Kolyada "was the Slavic god of peace and corresponded to the Roman Janus [. . . ] The temple of this god (or maybe goddess) was in Kiev . . . " (Sreznevskij 1817, pp. 19–20; italics in the original). Indeed, Sreznevsky published the carol as "a hymn to Kolyada". He did not directly assert that the goat was intended as a sacrifice to Kolyada, but implied it. Thus, the connection between the carol and pagan rites had already been suggested in Sreznevsky's publication; that is, Sakharov was following in a certain tradition which had formed even before he composed his carol. Finally, in the third place, scholars interpreted the fires as sacrificial fires, and the old man who "sharpens his steel knife" as a pagan priest.

Given Sakharov's record as a falsifier of folklore, there is no reason to doubt that it was he who created this carol. Oddly enough, there is not a single line in it that Sakharov

composed by himself: they are taken either from the carols of I.E. Sreznevsky or from versions of the folktale about Alyonushka and her brother Ivanushka. Despite the pseudo-folkloric nature of Sakharov's carol, several generations of historians and philologists have seen it as a description of a ritual performed by pagan Slavs in ancient times. The material considered in this article leads to the conclusion that the use of folklore as historical data requires extreme care and a close examination of the sources involved.

**Funding:** This research was funded by the Russian Foundation for Basic Research (grant number 20-012-00117).

**Institutional Review Board Statement:** I choose to exclude this statement if the study did not involve humans or animals.

**Informed Consent Statement:** I choose to exclude this statement as the study did not involve humans.

**Data Availability Statement:** I choose to exclude this statement as the study did not report any data.

**Acknowledgments:** I am thanking Marcus Levitt for his translation work.

**Conflicts of Interest:** The author declare no conflict of interest.

## Note

1. In the ceremony of "leading a goat" (*vozhdenie kozy ili kozla*) a group of mummers (often teenagers) sings a song about a goat. One of the mummers acts as the goat and the others lead him from house to house. The song usually tells how when a goat is out walking with its young children it is attacked by a wolf. The goat seems to have been killed but then comes back to life, as in the song minor harmonies change into major, joyful ones.

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
