# Peer review of "The Carol about the Pagan Rite of Sacrifice of a Goat and Its Interpretation in Russian Scholarship of the 19th to 20th Centuries"

_religions, doi:10.3390/rel12050366_

Round 1
Reviewer 1 Report
Russian words transcripted into Latin script should be written in italics, as well as the titles of books.

Author Response
I have italicized all titles and Russian words in accord with the reviewer's suggestions.
Reviewer 2 Report
OK, first and most concerning, about 200 or so words are reprinted verbatim (see lines listed below)
"Nationalism" should be added to keywords,
Folklorists have mostly moved on from the "fakelore" arguments of the 1970s and 80s because we realize that literary/oral divide was never a simple binary. I think this needs to be mentioned in a footnote. However, this doesn't weaken the article's argument because, in trying to reconstruct a (nationalist) past, it probably matters if something is folklore or fakelore. Should, however, be addressed.
26: Few who are not Russian Orthodox understand that their priests marry, so Sakharov's father being a priest might also warrant a footnote.
The Brothers Grimm rewrote some stories - see Zipes, Kamensky, Ellis or Haas. Precedent for blurring literary & oral distinction in European folklore.
80: extraneous "it"
195: add "by"
241: the incorporation of song from story is "unique." Maybe in this case, but not in general. Overlap of narrative and song is long acknowledged.
343-351: same lines used earlier in article; also 353-393 and 436-441 and 443-448 and 464-482
Without repetition, article would be too short. Repetition shouldn't be there. Did I in fact get the correct document, or is there a problem with the text that was sent to me?
Abrupt end - if (as the author establishes well) the carol has consistently been misinterpreted, what is the significance of this? Obviously people have written about a sacrificial tradition that probably didn't exist - so what? Does this show us something about Russian nationalism? About folklore and fakelore? About the dangers of using folklore texts to describe history? Tale the argument a step or two further (necessary to add words anyway because of the 200 or so that are repeated) in the conclusion.
Author Response
I have incorporated virtually all of the reviewer's suggestions. A couple did not work ( Line 181, change "Thus, Sakharov's assertion..." to "Sakharov asserts..." and Line 188-189, remove "The song in our collection fills in all omissions"). The recommendation that I move "the paragraph ( lines 690 - line 711) to the beginning of the text to be used as the introduction" was not possible in a mechanical way, since it wouldn't make sense, but I used some of it to rewrite the introduction.
Reviewer 3 Report
"The Carol about the Pagan Rite of Sacrifice of a Goat and Its Interpretation in Russian Scholarship of the 19th-20th Centuries examines the sources of a poem presented by Ivan Petrovich Sakharov, as an authentic example of a Russian folksong. After introducing the historical background of Sakharov and contextualizing the carol broadly, the author(s) undertake a studied comparison of Sakharov's text in relation to an earlier edition of the same carol published by Ivan Evseevich Sreznevsky. The author clearly traces the bulk of Sakharov's song, as a copy of Sreznevsky. The remainder of the carol is then traced to a Russian fairy tale, which is only implicit in Srenevsky's version. The author then demonstrates that the ritual sacrifice of the goat, the titular subject of the carol, is not a part of the original. Finally, the author shows how the transition of the implicit to explicit inclusion of the sacrificial scene has propagated through and erroneously influenced contemporary scholars, who perceive the carol as being relevant to Russian sacrificial practices.
The paper appears to present an incremental development in scholarship on Sakharov, moving from acceptance as an amateur historian of Russian folklore to a dishonest broker of the same, by presenting a more influenced examination of how his publications were created. However, editing and some structural modifications to the paper are needed to bring it to an acceptable level of quality.
- The introduction with the historical background does not clearly present the contents of the paper, if the abstract were absent. I recommend moving the paragraph (lines 690 - line 711) to the beginning of the text to be used as the introduction.
- In line 690, eliminate "thus" to being "Researchers..."
- Include a new section title between lines 711 and 712 - "Conclusion."
- Eliminate "To sum up," and start concluding paragraph with "Having..." at line 712.
- Eliminate sentence (719-720), beginning " Such contamination...: and paragraph break at start of 721.
- Throughout the paper, remove placeholder formatting and text content from the paper template provided by Religions.
- Line 46 change "typography" to "publishing house."
- Line 62 remove "eighteenth."
- Line 67 remove "of the technique."
- Line 70, end sentence at "...stylistic editing." and Begin new sentence "There are also..."
- Paragraph 75-80 - provide title of carol?
- Line 80, eleminate "it" at beginning of line.
- Change section title at line 81 to more clearly match content. Perhaps, "Origins of Sakharov's "Carol about...""
- For English translations of the carol, add in a hard return between original and English and between English and body of paper. It is difficult to read as presented.
- Are lines 128-134 a citation? Unclear if these are the author's words or Sakharov's. If not, move explanatory sentence "[It] is reported that..." (line 140-141) to line 130 after "...when performing songs." Eliminate "One the performance of the song" from the moved sentence.
- Line 134, change "...begins like this:" to "begins with the words:"
- Line 142 change "begins with the words:" to "begins:"
- Line 147 to 151, change to " Sakharov notes that the carol was recorded "from the words of Tula villagers." (Sakharov 1838, p. 163, note 18) In the section, "Variants of Christmastide Songs" Sakharov claims that only one version of the song exists, the one printed by Sreznevsky in the "Ukranian Bulletin" in 1817."
- Line 169, add "Sakharov states that, "Something..."
- Are lines 169-176 a citation? It is unclear.
- Line 181, change "Thus, Sakharov's assertion..." to "Sakharov asserts..."
- Line 188-189, remove "The song in our collection fills in all omissions."
- Line 195, swap "published" and "Sreznesky order.
- Line 195-916 Eliminate, "Firstly, we know" Capitalize "he".
- Line 196-197 Change to "...Bulletin," and Sakharov modified other texts of the oral tradition, such as Parikhin's..."
- Line 206-207, change to hard stop at Szrenevsky, capitalize "is..."
- Line 207 add "offer guidance with respect to the authenticity of Sreznevsky's text." Remove "We have found. Capitalize "three."
- Rewrite paragraph at lines 235-24, perhaps begin with something like, " Despite the coincidence of Sakharov and Sreznevsky's texts, the formers edits are not a reworking of the original. Instead, Sakharov amends the texts with elements of a fairy tale. In the fairy tale..."
- Line 244, after boy add "..., who has turned into a lamb."
- Line 249-250, add comma after length. Eliminate semicolon and start new sentence at "Sometimes."
- Note: throughout the text readability would be improved by splitting sentences combined by with a semicolon. In most of the uses of the semicolon present in the paper, the subordinate clause would be clearer if it stood on its own.
- Line 252, change from "of them:" to versions reads:"
- Line 322-323, edit sentence for clarity beginning with "But in the carol..."
- Lines 340-602 repeat prior portion of text. Remove.
- Line 603, is this the subsection title? Perhaps change to "3. Subsequent Use of Sakharov's Carol as Historical Evidence."
Author Response
I have incorporated virtually all of the reviewer's suggestions.
Round 2
Reviewer 3 Report
With the edits and adjustments, the essays reads much more clearly. It represents a very interesting account of the manner in which Sakharov approached the editing and extension of the carol, as well as tracing the subsequent misreading of Sakharov's reworking as historical fact. It would be interesting to determine if Sakharov's method of collaging separate sources in lieu of whole sale rewriting or extensively editing the text may have been what led to the later misinterpretations.